# Prognostic value of SPARC in hepatocellular carcinoma: A systematic review and meta-analysis

Xiaoyu Yang, Yunhong Xia[ORCID]*, Shuomin Wang, Chen Sun

Department of Oncology, The Fourth Affiliated Hospital, Anhui Medical University, Hefei, Anhui, China

* yhxia21@sina.com

**Data Availability Statement:** All relevant data are within the paper and its Supporting Information file.

**Funding:** The author(s) received no specific funding for this work.

## Abstract

### Objective

Hepatocellular carcinoma (HCC) is characterized by a high degree of malignancy, rapid proliferation of tumor cells, and early liver metastasis. Resistance to multiple drugs independent of the high expression of secreted protein acidic and rich in cysteine (SPARC) is associated with a high risk of recurrence and mortality. However, the prognostic value of SPARC in patients with HCC remains unclear. Therefore, we performed a meta-analysis to evaluate the relationship between the expression of SPARC and the prognosis of patients with HCC.

### Methods

We searched for relevant articles in the CNKI, PubMed, EMBASE, and Web of Science databases. The 95% confidence intervals (CIs) were calculated for combined overall survival (OS) and disease-free survival (DFS) to assess the prognostic value of expression of SPARC in patients with HCC.

### Results

In six of the studies, SPARC expression status was significantly associated with OS (combined hazard ratio [HR], 1.38; 95% CI, 1.0–1.82; Z = 2.27, P = 0.02) but not with DFS (combined HR, 0.79; 95% CI, 0.16–4.00, Z = 0.28, P = 0.78). Therefore, it cannot be assumed that upregulated SPARC expression has an effect on DFS in patients with HCC.

### Conclusion

Elevated SPARC expression is associated with a low survival rate but not with DFS in patients with HCC. Further studies are needed to confirm our conclusions.

### Registration

INPLASY registration number: INPLASY202180115. https://inplasy.com/inplasy-2021-8-0115/.

**Competing interests:** The authors have declared that no competing interests exist.

# Introduction

Hepatocellular carcinoma (HCC) is the sixth most prevalent cancer and the third leading cause of cancer-related deaths [1,2], with a rapidly increasing incidence and mortality rate [3]. HCC is usually diagnosed at an advanced and unresectable stage when only conventional treatment options can be used and has a median survival after diagnosis of 6–12 months [4,5]. Transarterial chemoembolization is currently the standard of care for HCC [6]. However, there is a significant disparity in the prognoses of individuals with unresectable HCC despite receiving the same treatment. Therefore, it is important to examine the prognostic indicators of HCC in clinical use.

Secreted protein acidic and rich in cysteine (SPARC), also known as osteonectin, was initially identified in bone and endothelial cells [7,8]. It is a 32–35-kDa multifunctional collagen or calcium-binding extracellular matrix glycoprotein belonging to a group of matricellular proteins encoded by genes located at 5q33.1 and consists of a single polypeptide (285 amino acids) comprising the following three biological structural domains: an acidic N-terminal domain, follistatin-like domain, and calcium-binding extracellular domain [9–11]. The human SPARC gene is expressed in numerous tissues and organs, including the bone marrow, whole blood, lymph node, thymus, brain, cerebellum, retina, heart, smooth muscle, skeletal muscle, spinal cord, intestine, colon, adipocytes, kidney, liver, pancreas, thyroid and salivary glands, skin, ovary, uterus, placenta, cervix, and prostate gland [12]. SPARC is a stromal cell glycoprotein that participates in the remodeling of the extracellular matrix and is involved in the development and progression of malignancies [10,13–19]. SPARC has been found to enhance tumorigenesis and metastasis and is associated with a poor prognosis [20–24], especially in pancreatic cancer [25,26], prostate cancer [27], and lung cancer [28]. SPARC is linked to the prognosis of HCC and can increase the proliferation and migration of tumor cells [5,29,30]. Furthermore, it helps HCC cells to acquire a stem cell morphology and promotes epithelial-mesenchymal transition, which is associated with tumor progression and metastasis [31].

Several researchers have evaluated the correlation between SPARC levels in hepatic stellate cells after activation and the prognosis of patients with HCC and found that independent high expression of SPARC can lead to high recurrence and mortality rates [32]. Meanwhile, it has been shown that the incidence of SPARC methylation in HCC tissue is much higher than that in non-tumor tissues and that patients without SPARC methylation have a higher postoperative overall survival (OS) rate than patients with SPARC methylation [33]. However, there are few relevant studies on the prognostic impact of SPARC in patients with HCC, and the predictive significance of SPARC in these patients is unknown. The aim of this meta-analysis was to analyze the prognostic significance of SPARC in patients with HCC to provide an evidence-based platform for future studies.

# Materials and methods

We registered this systematic review and meta-analysis with INPLASY (INPLASY202180115) and followed the Preferred Reporting Items for Systematic Reviews and Meta-Analysis (PRISMA) guidelines for meta-analyses [34].

## Database search strategy

We performed a systematic literature search of the CNKI, PubMed, EMBASE, and Web of Science databases from their inception to August 2021. The retrieval strategy was as follows: (1) PubMed: ("Carcinoma, Hepatocellular"[Mesh] OR "Carcinoma, Hepatocellular"[All Fields] OR "HCC"[All Fields] OR "liver cancer"[All Fields]) AND ("Osteonectin"[Mesh] OR

"Osteonectin"[All Fields] OR "SPARC"[All Fields] OR "Secreted protein acidic and cysteine rich"[All Fields]) AND ("Prognosis"[Mesh] OR "Prognosis"[All Fields] OR "Prognostic"[All Fields] OR "Survival Analysis"[Mesh] OR "Survival"); (2) EMBASE: ("hepatocellular carcinoma": ti, ab, kw OR hcc: ti, ab, kw OR "liver cancer": ti, ab, kw) AND (osteonectin OR sparc OR secreted) AND protein AND acidic AND cysteine AND rich AND (prognosis: ti, ab, kw OR prognostic: ti, ab, kw OR survival: ti, ab, kw); (3) Web of Science: TOPIC: (hepatocellular carcinoma OR HCC OR liver cancer) AND TOPIC: (Osteonectin OR SPARC OR Secreted protein acidic and cysteine rich) AND TOPIC: (Prognosis OR Prognostic OR Survival); (4) CNKI: keywords (hepatocellular carcinoma OR HCC OR liver cancer) and (Prognosis OR Prognostic OR Survival) in Chinese.

### Inclusion criteria

To be eligible for inclusion, the following criteria had to be fulfilled: (a) clinical study in patients with HCC; (b) SPARC expression in HCC measured using immunohistochemistry, quantitative real-time polymerase chain reaction, or western blotting; (c) association between SPARC expression and survival outcomes reported; (d) hazard ratio [HR] and 95% confidence interval [CI] for OS according to SPARC status either reported or able to be estimated from the relevant published data.

### Exclusion criteria

The following exclusion criteria were applied: (a) publication as a letter, editorial, abstract, review, case report, or expert opinion; (b) an in vitro or in vivo experiment; (c) HRs for OS with 95% CIs not reported and no Kaplan–Meier survival curves available; (d) duplicate article derived from an identical or overlapping patient population (only the most recent and/or complete one used); (e) inclusion of patients with a diagnosis of malignancies other than HCC.

### Literature screening

The literature search and screening were performed by two researchers working independently. The titles, abstracts, and keywords were read briefly; the complete text was then reviewed and selected based on the inclusion and exclusion criteria. Finally, inclusion was determined by cross-checking; if the two researchers did not agree on the selection of a particular article, a third qualified researcher was asked to adjudicate.

### Extraction of data

Information on the initial author, year of publication, number of study participants, sex ratio, and outcome indicators was collected and summarized using Excel 2019 software.

### Quality evaluation of the literature

The quality of the included studies was appraised by two authors using the risk of bias assessment technique in the Cochrane Handbook for Systematic Reviews of Treatments 5.1.0, which is used to assess the risk of publication bias. Investigators and outcome assessors were blinded to all information on random sequence generation and allocation concealment. All articles were evaluated for completeness of outcome data and selective reporting of research outcomes.

## Statistical methods

Stata 11.2 and RevMan 5.3 software were used to evaluate the relevant literature. All outcome indicators were continuous variables, and the data are expressed as the odds ratio (OR) with the 95% CI. Heterogeneity between the included studies was estimated. The Q test was performed to determine the heterogeneity of the I2 response. A P-value > 0.1 or an I2 value < 50% indicated statistically significant homogeneity, and a fixed-effects model was used to conduct the meta-analysis. A P-value < 0.1 or an I2 value > 50% indicated statistically significant heterogeneity, and a random-effects model was used with subgroup analysis to investigate the source of heterogeneity. The combined effect size test indicated a statistically significant difference at P ≤ 0.05.

## Results

### Search results

The initial systematic search identified 1235 studies. After eliminating duplicates, 195 articles remained. Based on the title and abstract screening, 72 irrelevant articles were excluded, and 45 were further screened for assessment of eligibility. Thirty-nine studies that did not meet our inclusion criteria were excluded, leaving six [32,33,35–38] for inclusion in the meta-analysis. Fig 1 shows the literature search and article selection processes.

### Study characteristics and quality assessment

This meta-analysis included all studies published in the English language between 2009 and 2020. The studies included a total of 678 patients with HCC. Background data, including sex ratio, mean age, and duration of disease, were comparable between the study and control groups. The patient sample size ranged from 60 to 200 in the six studies. In all studies, SPARC was detected in tumor tissues, mesenchymal cells, or cancer cells by immunohistochemistry or quantitative real-time polymerase chain reaction. Stratification of SPARC expression differed between the studies. HRs for reported or estimated OS and disease-free survival (DFS) with 95% CIs were included. SPARC was mentioned as a predictor of poor prognosis in four studies, and only one [38] found that it had no effect on OS. SPARC was found to be a good predictor of outcomes in one study [35]. The patient characteristics are summarized in Table 1. Huang et al. used quantitative polymerase chain reaction (qPCR) to determine SPARC levels in 120 samples. OS and DFS were recorded [35]. Liu et al. recorded OS and used immunohistochemistry (IHC) to determine SPARC levels in 89 samples [36]. Yang et al. used IHC to determine SPARC levels in 79 samples. OS and DFS were recorded [37]. Ju et al. used qPCR to determine SPARC levels in 130 samples. OS and relapse-free survival (RFS) were recorded [32]. Zhang et al. recorded OS and used methylmion specific polymerase chain reaction (MSP) to determine SPARC levels in 60 samples [33]. Darweesh et al. recorded OS and used qPCR to determine SPARC levels in 200 samples [38].

Five reports clearly included the method of group assignment. Huang et al. used paired paraffin-embedded samples from patients with HCC who had undergone liver resection. Adjacent non-cancerous liver tissue samples were selected as the control group [35]. Liu el al. performed immunohistochemical assays in hepatic tissues collected from patients with HCC (study group) and healthy individuals (control group) [36]. Yang et al. selected patients with primary liver cancer who underwent surgery as the study group and tumor adjacent liver tissues as the control group [37]. Ju et al. selected patients with HCC matching the following criteria to form the study group: (1) pathologically diagnosed HCC; (2) no anticancer treatment or distant metastases before surgery; and (3) history of HCC resection, defined as macroscopically

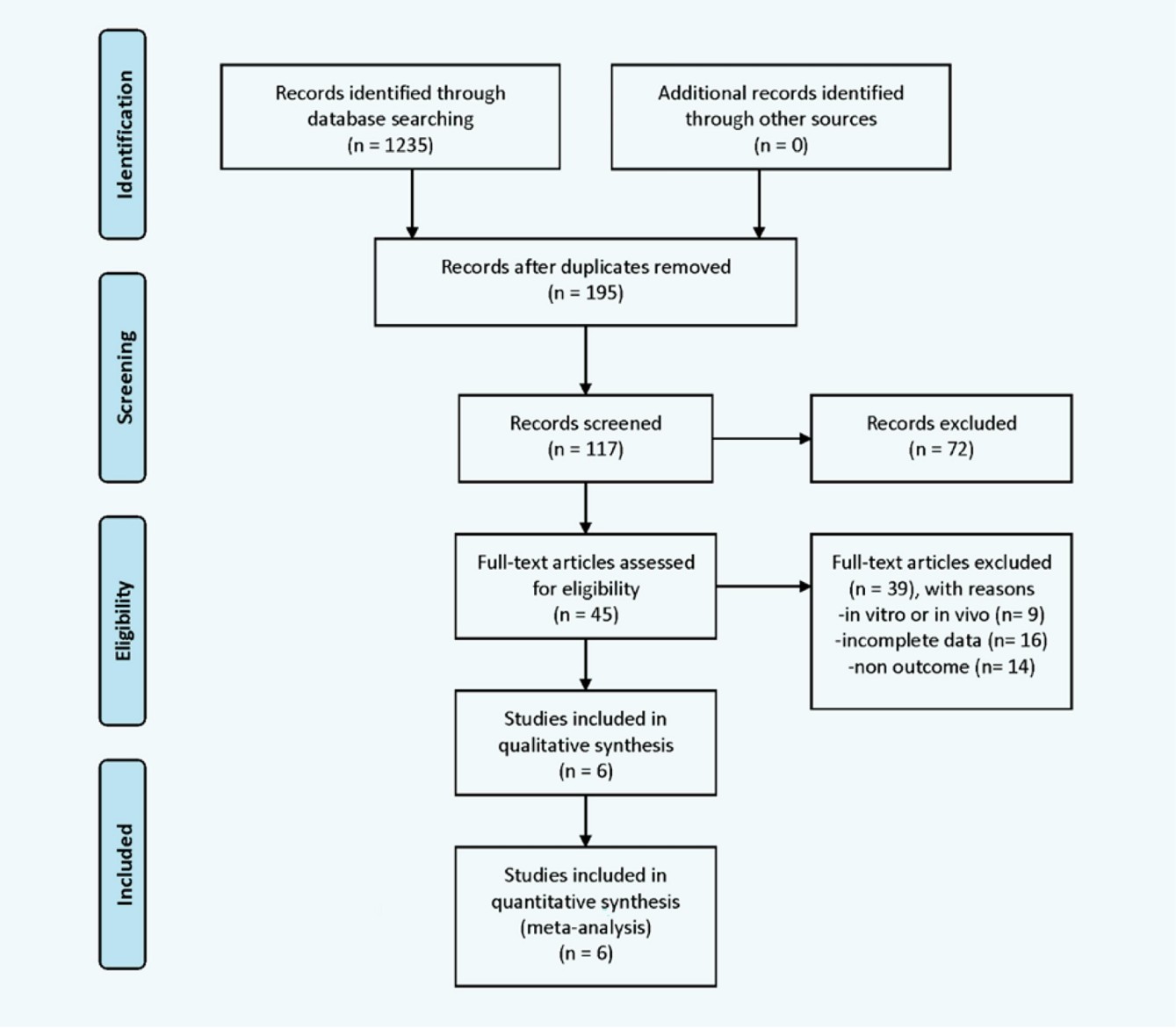

**Fig 1. Flow chart showing the process used to select articles for meta-analysis.**

complete removal of the tumors. Meanwhile, a normal liver tissue pool from 10 healthy liver donors was assigned as the control group [32]. Zhang et al. selected HCC tissues as the study group and nontumorous tissues as the control group [33]. Darweesh et al. selected patients with HCC as the study group and healthy volunteers as the control group. Healthy paticipants were those who had normal liver enzyme levels and functions, negative results on viral hepatitis screening, and normal abdominal ultrasound scans [38]. Six did not mention blinding in the outcome analysis. Specific grouping information was not clearly described; only one study was evaluated as having a high risk of bias, and the degree of assignment bias was unknown in one study. Three studies did not specify whether testing was performed using blinded principles, and the amount of missing data was not specified in one article. Testing bias was identified in three studies. Some of the included studies did not describe the staging of HCC. The quality assessment results are shown in Fig 2.

**Table 1. Background characteristics of the study population.**

| First author | Country | Method | Subjects | Sex M/F | Cell type/ location | Study group | Control group | SPARC + | SPARC - | Outcome | Obtainment |
|---|---|---|---|---|---|---|---|---|---|---|---|
| Huang, et al. 2017 [35] | China | qPCR | 120 | 110/10 | Tumor | HCC tumor tissues | Tumor adjacent liver tissues | 71 | 49 | OSDFS | Multivariate |
| Liu 2020 [36] | China | IHC | 89 | 36/7 | Stroma | Patients with HCC | Healthy controls | 11 | 78 | OS | Multivariate |
| Yang 2018 [37] | China | IHC | 79 | 38/16 | Tumor | Patients with primary liver cancer | Tumor adjacent liver tissues | 35 | 44 | OSDFS | Multivariate |
| Ju 2009 [32] | China | qPCR | 130 | 112/18 | Tumor | Patients with HCC | Healthy liver donors | 85 | 45 | OSRFS | Multivariate |
| Zhang 2012 [33] | China | MSP | 60 | 9/51 | Tumor | HCC tumor tissues | Nontumorous liver tissues | 21 | 39 | OS | Multivariate |
| Darweesh 2018 [38] | Egypt | qPCR | 200 | 160/40 | Tumor | Patients with HCC | Healthy controls | NA | NA | OS | Multivariate |

## SPARC and overall survival

Six studies showed an association between SPARC expression and OS. There was considerable heterogeneity in OS in patients with HCC between the studies ($\chi^2$ = 26.96, P<0.001; $I^2$ = 81%). The random-effects model was used for the analysis because the $I^2$ value was >50%, and the combined HR for the six studies was 1.38 (95% CI 1.04–1.82; Z = 2.27, P = 0.02), indicating that upregulated expression of SPARC was significantly associated with OS in patients with HCC (Fig 3).

Two studies revealed the relationship between SPARC expression and DFS, as shown in Fig 4. There was significant heterogeneity in the data ($\tau^2$ = 1.23; $\chi^2$ = 26.96, P = 0.002; $I^2$ = 90%). Therefore, a random-effects model was used for analysis; the combined HR for the two studies was 0.79 (95% CI, 0.16–4.00; Z = 0.28, P = 0.78), which did not indicate a statistically significant difference. Therefore, upregulated SPARC expression did not appear to affect DFS in patients with HCC.

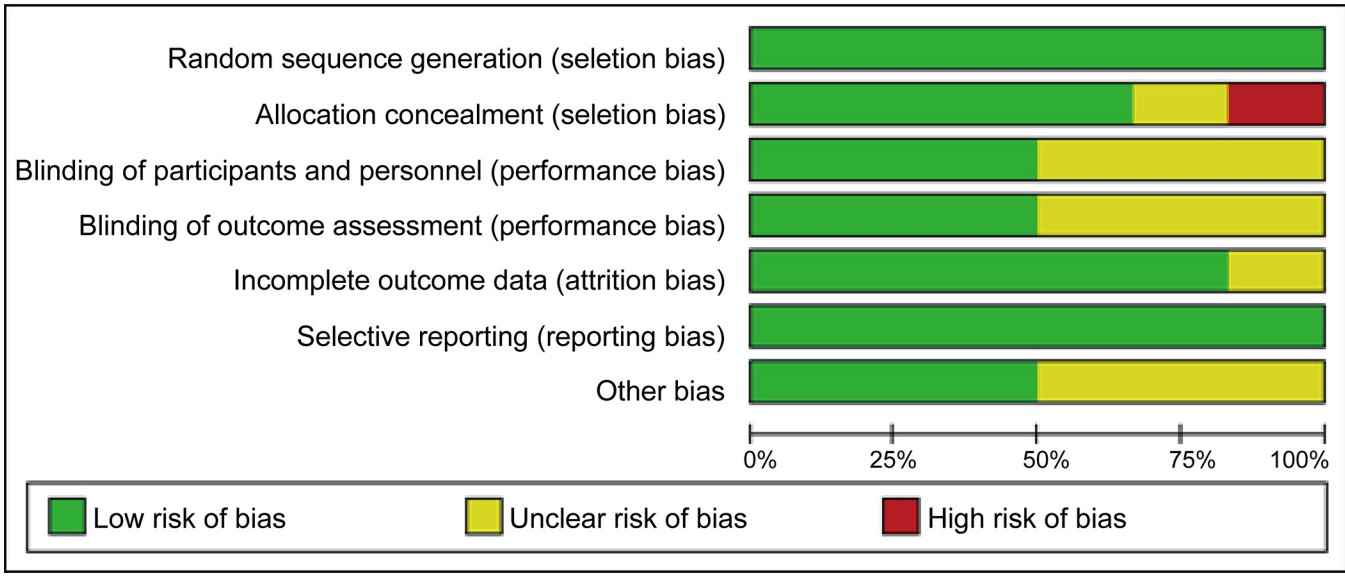

**Fig 2. Quality assessment chart.**

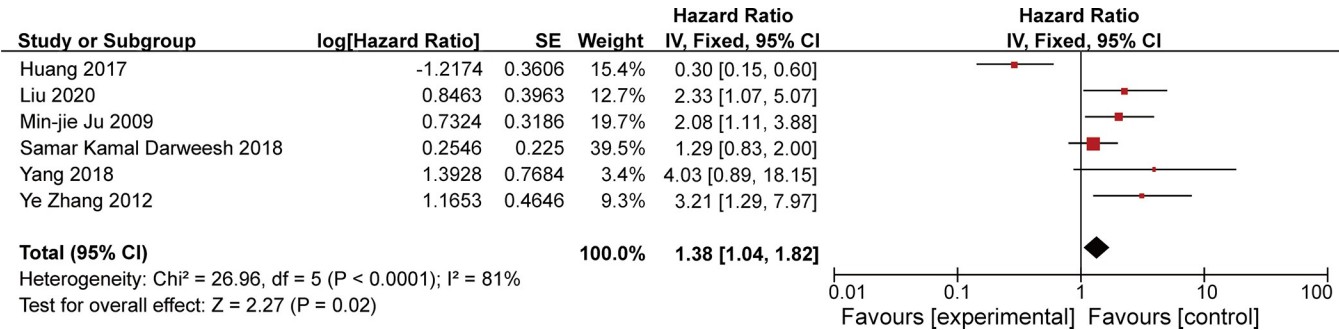

**Fig 3. Forest plot showing the correlation between change in overall survival and secreted protein acidic and rich in cysteine expression in patients with hepatocellular carcinoma.**

Funnel plots (Figs 5 and 6) constructed to assess publication bias in the studies that included both OS and DFS showed an asymmetric distribution, indicating considerable publication bias.

## Discussion

SPARC is attracting increasing interest as a potential prognostic biomarker in patients with cancer [10,12,39–43]. However, the link between the SPARC expression profile and patient survival is a matter of debate and seems to depend on the type of tumor [44,45]. There have been many studies on SPARC expression in patients with pancreatic [46–48], prostate [27,49], lung [28,50], breast [27,51] and other tumors in terms of carcinogenesis, metastasis, and prognosis [52,53]. However, although upregulated SPARC expression is thought to be associated with a favorable outcome [54], the functional role of SPARC in cancer varies according to tumor type and tissue environment [55]. SPARC expression has been demonstrated to both promote and inhibit various forms of tumor cell activity. Although the sample size in SPARC studies in HCC is very small, several studies have shown that SPARC can boost proliferation and migration of tumor cells and that it may be associated with the prognosis of HCC [5,31,56]. However, until now, there has been no meta-analysis of studies that have suggested the prognostic significance of SPARC in HCC.

Our meta-analysis of six eligible studies including a total of 678 patients is the first to systematically evaluate the role of SPARC in the prognosis of HCC. We found that patients with HCC who had high SPARC expression had a lower OS rate than patients without HCC did. When the HR value for DFS was examined, there was no relationship between expression of SPARC in patients with HCC and DFS. However, only two of the included studies included DFS evaluation and had conflicting results, which is presumably the reason for this negative

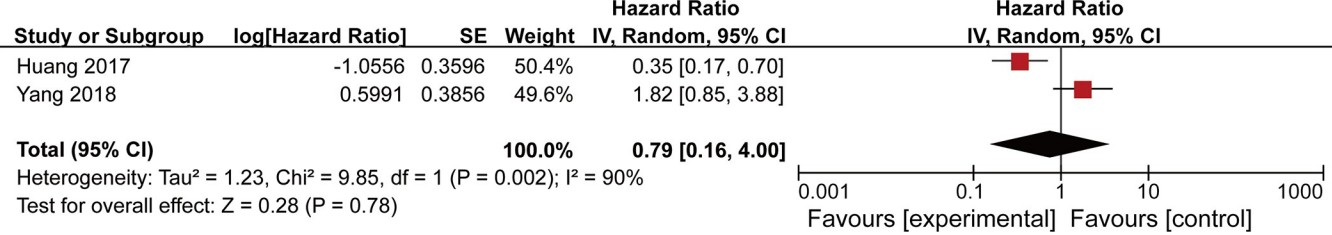

**Fig 4. Forest plot showing the correlation between disease-free survival and secreted protein acidic and rich in cysteine expression in patients with hepatocellular carcinoma.**

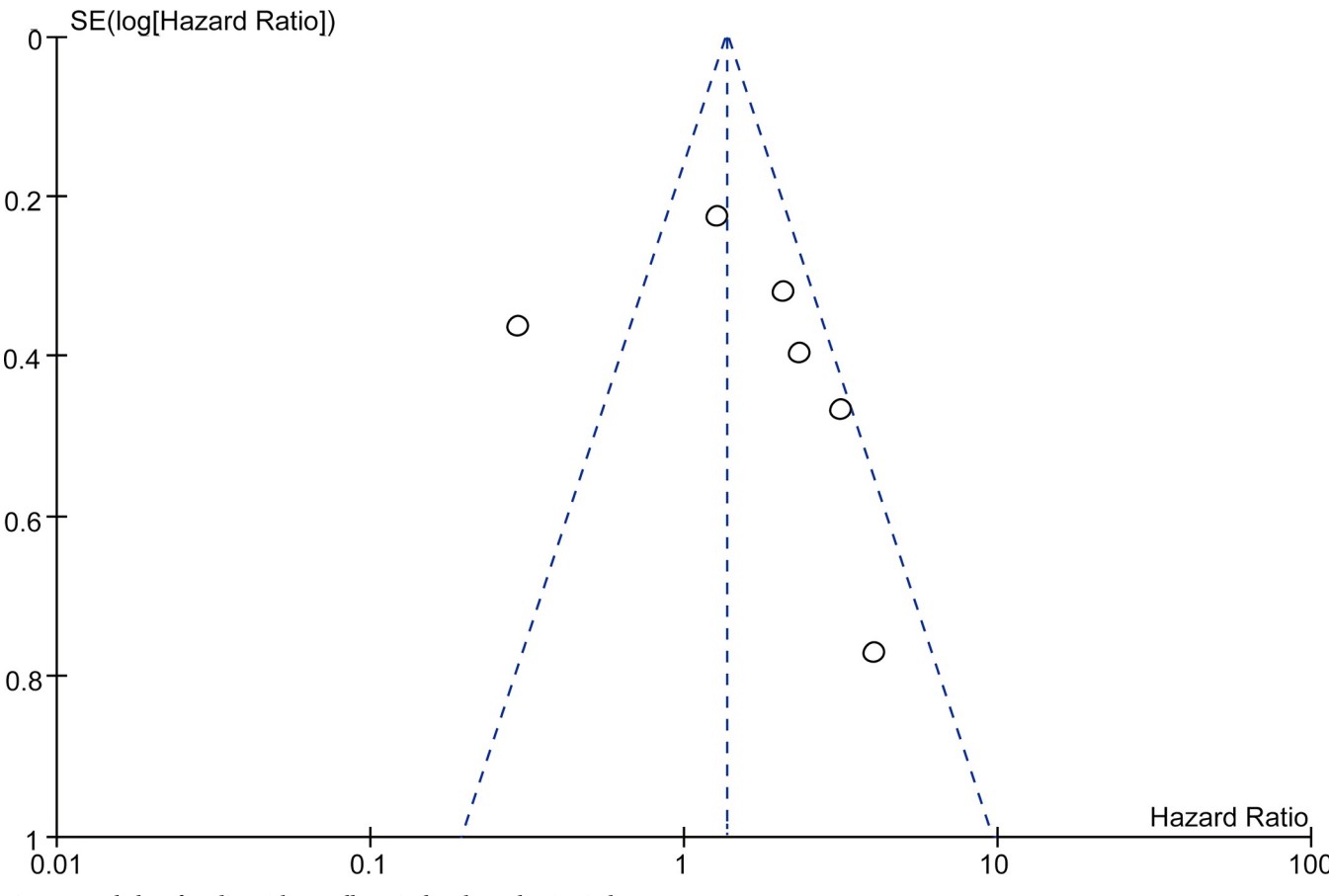

**Fig 5. Funnel plot of studies with overall survival as the evaluation index.**

finding, requiring further study. We also created funnel plots for OS and DFS to analyze bias and found them to be asymmetric, indicating that our results were unstable.

Other researchers have identified abnormal methylation of SPARC in some tumor cell lines and that SPARC methylation induces gene silencing and inhibition of tumor activity [57], whereas demethylating agents can change methylation status and restore gene expression. Therefore, upregulated expression of SPARC in patients with HCC is most likely attributable to methylation. SPARC methylation is more common in HCC tissue [37], and patients with SPARC methylation have a low overall postoperative survival rate. Moreover, multiple significant molecular processes involving SPARC have been found in malignant tumors in both the extracellular matrix and tumor microenvironment, including regulation of modulation, anti-adhesion, apoptosis, growth, migration, and invasion [53]. Furthermore, the tumor suppressor KLF4 decreases tumor invasion by downregulating the expression of SPARC [58]. These findings are consistent with the theory that high SPARC expression is associated with a poor prognosis in patients with HCC.

This research has some limitations. First, the sample size was limited to 678 patients, most of whom were from health centers or hospitals with adequate follow-up, and the quality of the data was variable, which may have affected our results. Second, given that most of the studies were retrospective, selection bias and information bias were inevitable. We identified some risk of bias (Fig 2), especially for allocation concealment. Third, although there is a link

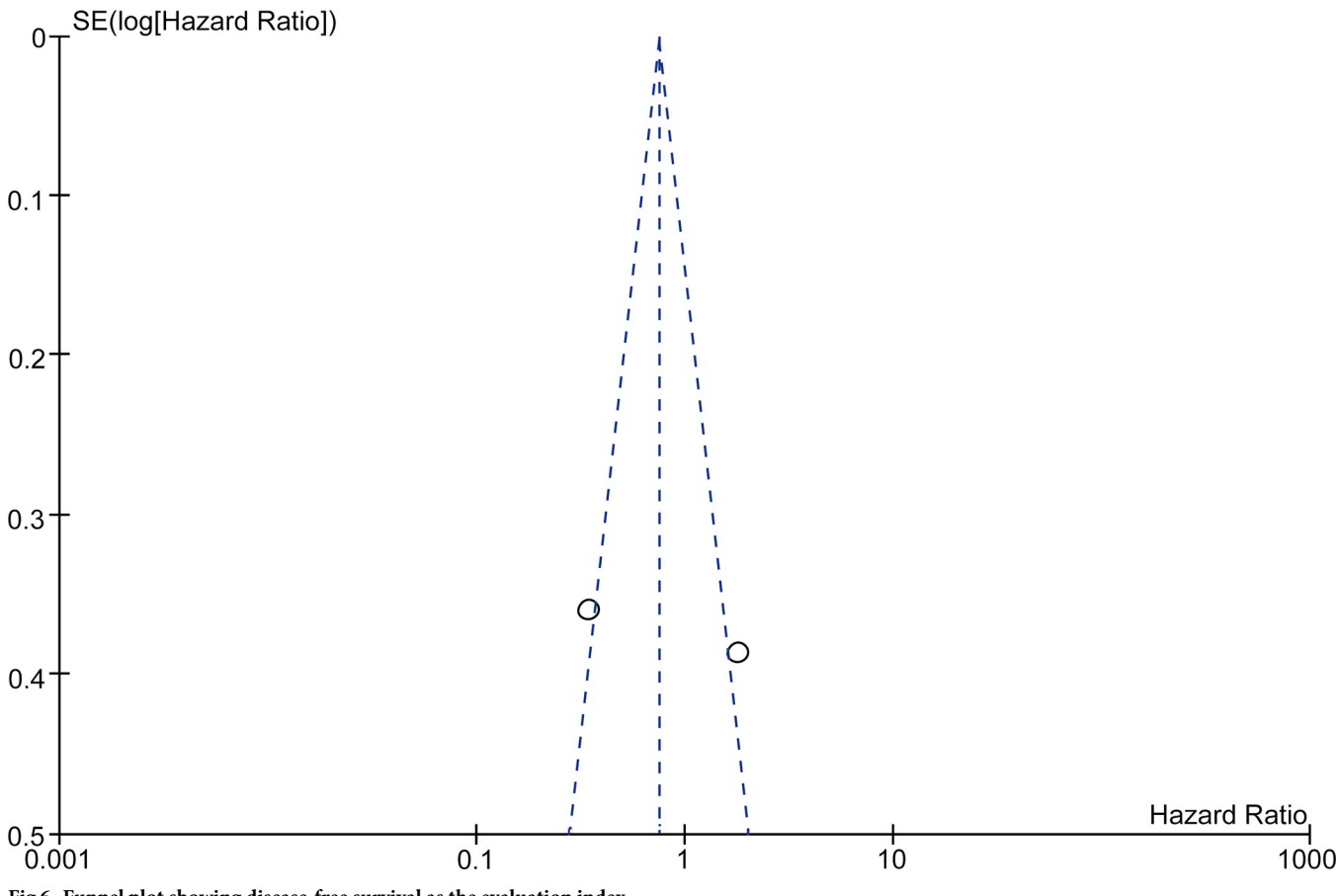

**Fig 6. Funnel plot showing disease-free survival as the evaluation index.**

between SPARC expression and tumor stage, the TNM (tumor node metastasis) staging system was not used in any of the eligible studies. Therefore, the inclusion of patients with various disease stages may have had an impact on our findings, including for heterogeneity. Finally, we only included studies published in English, which may have resulted in the exclusion of relevant studies published in other languages. Furthermore, the number of relevant studies that have not been published is unknown. Therefore, further research is needed to confirm our present findings regarding the prognostic value of SPARC in patients with HCC.

## Conclusion

This comprehensive review and meta-analysis found an association between high SPARC expression and a poor prognosis in patients with HCC. However, this association did not extend to DFS, possibly because DFS was rarely included in the eligible studies. Therefore, more research is needed to confirm our findings and investigate the molecular mechanisms and pathways that influence DFS.

## Supporting information

**S1 File. PRISMA 2010 checklist.**
(DOCX)

## Author Contributions

**Conceptualization:** Xiaoyu Yang, Yunhong Xia, Shuomin Wang, Chen Sun.

**Data curation:** Xiaoyu Yang, Yunhong Xia.

**Formal analysis:** Xiaoyu Yang, Yunhong Xia.

**Investigation:** Xiaoyu Yang.

**Methodology:** Xiaoyu Yang.

**Project administration:** Xiaoyu Yang.

**Resources:** Xiaoyu Yang.

**Software:** Xiaoyu Yang.

**Supervision:** Xiaoyu Yang, Yunhong Xia.

**Validation:** Xiaoyu Yang.

**Visualization:** Xiaoyu Yang.

**Writing – original draft:** Xiaoyu Yang.

**Writing – review & editing:** Xiaoyu Yang, Yunhong Xia, Shuomin Wang, Chen Sun.

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
