## [Decision Letter · Decision Letter 0]

24 Jan 2022

PONE-D-21-33504Prognostic value of SPARC in hepatocellular carcinoma: a systematic review and meta-analysisPLOS ONE

Dear Dr. Xia,

Thank you for submitting your manuscript to PLOS ONE. After careful consideration, we feel that it has merit but does not fully meet PLOS ONE’s publication criteria as it currently stands. The reviewer(s) recommended **major revision **to your manuscript. Therefore, we invite you to submit a revised version of the manuscript that addresses the points raised during the review process.Please submit your revised manuscript by 28th February 2022. If you will need more time than this to complete your revisions, please reply to this message or contact the journal office at plosone@plos.org. Please include the following items when submitting your revised manuscript:A rebuttal letter that responds to each point raised by the academic editor and reviewer(s). You should upload this letter as a separate file labeled 'Response to Reviewers'.A marked-up copy of your manuscript that highlights changes made to the original version. You should upload this as a separate file labeled 'Revised Manuscript with Track Changes'.An unmarked version of your revised paper without tracked changes. You should upload this as a separate file labeled 'Manuscript'.

We look forward to receiving your revised manuscript.

Kind regards,

Raj Kumar Koiri

Academic Editor

PLOS ONE

Journal Requirements:

2. If you spot an unpublished reference (e.g. a reference that states “unpublished”, “in preparation” or “submitted”), ask the authors to amend the reference. However, this is not one of our normal checks and we should not be looking for any unpublished references. If you come across this issue, use the send-back below and amend the items in red as necessary.

**Reviewer Comments to the Author**

Reviewer #1: 1&2&3. What are the justifications for the exclusion criteria? For instance, why were similar studies excluded, how was their ‘similarity’ measured, and how did the authors decide which study among similar studies to include?

The ‘Cochrane handbook for systematic reviews of treatments’ (please cite this resource/include it in the references) was used to evaluate publication bias; however, the criteria is different from what is described in the handbook. For instance, what does ‘Other bias’ mean?

The conclusions regarding overall survival and disease-free survival are supported by only 6 and 2 studies, respectively. Due to the small number of studies reviewed and as mentioned in the article, no comprehensive evidence is presented to support the conclusions.

4.A number of sections in this article contain grammatical errors, making it difficult to understand the text. Acronym definitions are not consistent in terms of letter case and whether the acronym is included in parenthesis or the other way around.

The ‘Country’ column in Table 1 seems to be the author’s affiliation, not the country.

Would it be possible to include figures of higher quality? It’s difficult to read the blurred text in figures.

Reviewer #2: The authors are presenting a novel meta-analysis to analyze the predictive significance of SPARC in Hepatocellular Carcinoma patients.

Regardless of the small sample size, the data presented was novel and can be utilized to build future research in trying to pinpoint the association between SPARC and HCC

I do not have any comments for the authors

Reviewer #3: The current manuscript presented a review and meta-analysis to evaluate the reported evidences in scientific domain in order to understand SPARC expression and the prognosis of patients with HCC. The attempt is worth trying but investigators left noticeable methodological flaws in conduct of this meta-analysis/reporting - making results less reliable. Some of Points are mentioned below:

Authors refer present work as Network Meta analysis which is incorrect. (L99)

Authors have not provided Search Strategy Algorithms. The described search strategy in manuscript is of poor quality.

Systematic Review part of manuscript is incomplete, and there is no qualitative synthesis of evidence based on previously reported work.

Authors report that present work included the studies between 2019 and 2020 (L173), however, Table 1 does not comply with that criteria. The manuscript has many such contradictions- seems authors need to verify their data and description text.

Reason for the selection of Random Effects Model is insufficient. Results are insufficiently described which makes difficult to follow the conclusion and discussion part of manuscript.

---

## [Author Response · Author response to Decision Letter 0]

26 Feb 2022

Reviewer #1:

1. 1&2&3. What are the justifications for the exclusion criteria? For instance, why were similar studies excluded, how was their ‘similarity’ measured, and how did the authors decide which study among similar studies to include?

Response: Thank you very much for your constructive and helpful suggestions. In this paper, similarity refers to different articles using the same case. In order to eliminate misunderstandings, we corrected the manuscript again. Please see Line 106-119.

2. The ‘Cochrane handbook for systematic reviews of treatments’ (please cite this resource/include it in the references) was used to evaluate publication bias; however, the criteria is different from what is described in the handbook. For instance, what does ‘Other bias’ mean?

Response: Thank you very much. The wrong statement has been corrected. The right quality evaluation of the literature is as flowed: The quality of the included studies was appraised by two authors using the risk bias assessment technique in the Cochrane Handbook for Systematic Reviews of Treatments 5.1.0, which assesses the risk of publication bias. Investigators and outcome assessors were blinded to all information on random sequence generation and allocation concealment. All articles were evaluated for completeness of outcomes data and selective reporting of research outcomes. Please see Line 130-136.

3. The conclusions regarding overall survival and disease-free survival are supported by only 6 and 2 studies, respectively. Due to the small number of studies reviewed and as mentioned in the article, no comprehensive evidence is presented to support the conclusions.

Response: Thanks a lot. Although only six studies were included in terms of overall survival, data analysis showed that increased expression of SPARC was significantly associated with OS differences in HCC patients. Therefore, the expression of SPARC has an impact on HCC patients, but more high-quality studies are needed to confirm this outcome in the future. This belongs to the limitations of this meta-analysis. Please see Line 256-268.

4. A number of sections in this article contain grammatical errors, making it difficult to understand the text. Acronym definitions are not consistent in terms of letter case and whether the acronym is included in parenthesis or the other way around.

The ‘Country’ column in Table 1 seems to be the author’s affiliation, not the country.

Response: Thank you very much for your helpful suggestions. We invited native English speakers to polish the language of this paper. In addition, Table 1 has been checked and the wrong words have been removed in revised version. 

5. Would it be possible to include figures of higher quality? It’s difficult to read the blurred text in figures.

Response: Thank you for your valuable suggestions. All figures have been checked. The figures of higher quality have been added instead of the old figures. Please see the figure files. 

Reviewer #2: The authors are presenting a novel meta-analysis to analyze the predictive significance of SPARC in Hepatocellular Carcinoma patients.

Regardless of the small sample size, the data presented was novel and can be utilized to build future research in trying to pinpoint the association between SPARC and HCC

I do not have any comments for the authors.

Response: Thank you very much for your constructive and helpful suggestions.

Reviewer #3: The current manuscript presented a review and meta-analysis to evaluate the reported evidences in scientific domain in order to understand SPARC expression and the prognosis of patients with HCC. The attempt is worth trying but investigators left noticeable methodological flaws in conduct of this meta-analysis/reporting - making results less reliable. Some of Points are mentioned below:

1. Authors refer present work as Network Meta analysis which is incorrect. (L99)

Response: Thank you very much for your helpful suggestions. The word “network meta-analyses” has been modified to “meta-analyses”. Please see Line 88.

2. Authors have not provided Search Strategy Algorithms. The described search strategy in manuscript is of poor quality.

Response: Thank you very much for your constructive comments. The detailed search strategy has been added into the revised manuscript. Please see Line 90-105.

The retrieval strategy was as follows: (1) PubMed: (“Carcinoma, Hepatocellular”[Mesh] OR “Carcinoma, Hepatocellular”[All Fields] OR “HCC”[All Fields] OR “liver cancer”[All Fields]) AND (“Osteonectin”[Mesh] OR “Osteonectin”[All Fields] OR “SPARC”[All Fields] OR “Secreted protein acidic and cysteine rich”[All Fields]) AND (“Prognosis”[Mesh] OR “Prognosis”[All Fields] OR “Prognostic”[All Fields] OR “Survival Analysis”[Mesh] OR “Survival”); (2) Embase: (“hepatocellular carcinoma”: ti, ab, kw OR hcc: ti, ab, kw OR “liver cancer”: ti, ab, kw) AND (osteonectin OR sparc OR secreted) AND protein AND acidic AND cysteine AND rich AND (prognosis: ti, ab, kw OR prognostic: ti, ab, kw OR survival: ti, ab, kw); (3) Web of Science: TOPIC: (hepatocellular carcinoma OR HCC OR liver cancer) AND TOPIC: (Osteonectin OR SPARC OR Secreted protein acidic and cysteine rich) AND TOPIC: (Prognosis OR Prognostic OR Survival); (4) CNKI: keywords (hepatocellular carcinoma OR HCC OR liver cancer) and (Prognosis OR Prognostic OR Survival) in Chinese.

3. Systematic Review part of manuscript is incomplete, and there is no qualitative synthesis of evidence based on previously reported work.

Response: Thank you very much. Systematic Review part of manuscript has been modified (Line 149-155). The initial systematic search identified 1235 studies. After eliminating duplicates, 195 articles remained. Based on the title and abstract screening, 72 irrelevant articles were excluded, and 45 were further screened for assessment of eligibility. Thirty-nine studies that did not meet our inclusion criteria were excluded, leaving six for inclusion in the meta-analysis. Fig 1 shows the literature search and article selection processes.

4. Authors report that present work included the studies between 2019 and 2020 (L173), however, Table 1 does not comply with that criteria. The manuscript has many such contradictions- seems authors need to verify their data and description text.

Response: Thank you very much for your kind comments. The wrong word “2019” has been modified to “2009”. Please see Line 161.

5. Reason for the selection of Random Effects Model is insufficient. Results are insufficiently described which makes difficult to follow the conclusion and discussion part of manuscript.

Response: Thank you very much. In statistical analysis, the Q test was used to determine the size of I2 response heterogeneity: if P >0.1 and I2<50%, it indicates statistical homogeneity, and a fixed-effect model was used to conduct the meta-analysis; if P<0.1 or I2>50%, it indicates statistical heterogeneity, and a random effect model was used to conduct the meta-analysis and subgroup analysis to investigate the source of heterogeneity [1, 2]. Therefore, in this study, a randomized controlled model was selected due to I2>50%. Meanwhile, the confusing sentence has been removed in revised version and the right reason has been added into the revised manuscript. Please see Line 190-191.

References

1. Sohouli MH, Fatahi S, Lari A, Lotfi M, Seifishahpar M, Gaman MA, et al. The effect of paleolithic diet on glucose metabolism and lipid profile among patients with metabolic disorders: a systematic review and meta-analysis of randomized controlled trials. Crit Rev Food Sci Nutr. 2021:1-12. Epub 2021/01/26. doi: 10.1080/10408398.2021.1876625. PubMed PMID: 33492173.

2. Schwingshackl L, Zahringer J, Nitschke K, Torbahn G, Lohner S, Kuhn T, et al. Impact of intermittent energy restriction on anthropometric outcomes and intermediate disease markers in patients with overweight and obesity: systematic review and meta-analyses. Crit Rev Food Sci Nutr. 2021;61(8):1293-304. Epub 2020/05/05. doi: 10.1080/10408398.2020.1757616. PubMed PMID: 32363896.

---

## [Decision Letter · Decision Letter 1]

9 May 2022

PONE-D-21-33504R1Prognostic value of SPARC in hepatocellular carcinoma: a systematic review and meta-analysisPLOS ONE

Dear Dr. Xia,

Thank you for submitting your manuscript to PLOS ONE. After careful consideration, we feel that it has merit but does not fully meet PLOS ONE’s publication criteria as it currently stands. Therefore, we invite you to submit a revised version of the manuscript that addresses the points raised during the review process.

We look forward to receiving your revised manuscript.

Kind regards,

Raj Kumar Koiri

Academic Editor

PLOS ONE

Journal Requirements:

Reviewers' comments:

Reviewer's Responses to Questions

**Comments to the Author**

1. If the authors have adequately addressed your comments raised in a previous round of review and you feel that this manuscript is now acceptable for publication, you may indicate that here to bypass the “Comments to the Author” section, enter your conflict of interest statement in the “Confidential to Editor” section, and submit your "Accept" recommendation.

Reviewer #1: All comments have been addressed

Reviewer #2: All comments have been addressed

Reviewer #3: All comments have been addressed

Reviewer #4: All comments have been addressed

2. Is the manuscript technically sound, and do the data support the conclusions?

Reviewer #1: (No Response)

Reviewer #2: Yes

Reviewer #3: Yes

Reviewer #4: Yes

3. Has the statistical analysis been performed appropriately and rigorously? 

Reviewer #1: (No Response)

Reviewer #2: Yes

Reviewer #3: Yes

Reviewer #4: Yes

4. Have the authors made all data underlying the findings in their manuscript fully available?

Reviewer #1: (No Response)

Reviewer #2: Yes

Reviewer #3: Yes

Reviewer #4: Yes

5. Is the manuscript presented in an intelligible fashion and written in standard English?

Reviewer #1: (No Response)

Reviewer #2: Yes

Reviewer #3: Yes

Reviewer #4: Yes

6. Review Comments to the Author

Reviewer #1: (No Response)

Reviewer #2: I have no further comments for the authors.

Reviewer #3: Thank you for providing the additional information in respect of comments. The Queries for the manuscript have been addressed well.

Reviewer #4: In the manuscript titled “Prognostic value of SPARC in hepatocellular carcinoma: a systematic review and meta-analysis” the authors, with the help of data mining evaluated the SPARC expression and posed SPARC as a prognostic marker in hepatocellular carcinoma patients. This is novel in terms of the project plan without any wet laboratory experiments but only with the help of previous work and literature search with certain inclusion and exclusion criteria. The current manuscript is well written with some minor english grammar and typo mistakes. The pointwise comments are as follows:

1. Though the manuscript is well organized but is not easy to understand what is the goal/aim of this study.

2. When there are many other proteins (PIMD 33312759, 34591393, 32439236, 28631558) studied in the HCC why SPARC has been chosen, and what is the rationale behind it?

3. It would be better if the author had defined, what is “overall survival” and what is “disease-free survival”.

4. The entire manuscript is based on data mining and six paper has been selected which is a very small number.

5. The total sample size based upon the 6 papers is 678, according to table 1 (120+89+79+130+60+200 = 678), why and how it is mentioned 698 in line 233 page 13 and line 256 page 14.

6. This manuscript is not a regular research article, many things in the manuscript have to be defined, like “method of group assignment,” (line 177 page 10) and “blinding in the outcome analysis” (line 178 page 10).

7. Instead of presenting in the tabular form, it would be better if all the 6 studies were described with a separate heading including the sample size, methodology used, and putative outcome.

8. Some sentences are incoherent, e.g. “The p53/p21Cip1/Waf1 pathway enhances the effects of radiotherapy and chemotherapy” what does this sentence mean, and how it is correlated with the study?

9. While talking about the limitations of the paper, on page no 14, it is mentioned “other types of biases” in line 260., It is better to describe the “other biases” or just delete this line.

10. What does TNM stand for in line 262, page 14?

7. PLOS authors have the option to publish the peer review history of their article (what does this mean?). If published, this will include your full peer review and any attached files.

Reviewer #1: No

Reviewer #2: No

Reviewer #3: No

Reviewer #4: **Yes: **Ratnakar Tripathi

---

## [Author Response · Author response to Decision Letter 1]

15 Jun 2022

Thank you very much for your excellent work and sophisticated comments.

---

## [Decision Letter · Decision Letter 2]

8 Aug 2022

Prognostic value of SPARC in hepatocellular carcinoma: a systematic review and meta-analysis

PONE-D-21-33504R2

Dear Dr. Xia,

We’re pleased to inform you that your manuscript has been judged scientifically suitable for publication and will be formally accepted for publication once it meets all outstanding technical requirements.

Kind regards,

Raj Kumar Koiri

Academic Editor

PLOS ONE

Additional Editor Comments (optional):

The authors have satisfactorily answered the reviewers comment and the manuscript can be accepted.

Reviewers' comments:

Reviewer's Responses to Questions

**Comments to the Author**

1. If the authors have adequately addressed your comments raised in a previous round of review and you feel that this manuscript is now acceptable for publication, you may indicate that here to bypass the “Comments to the Author” section, enter your conflict of interest statement in the “Confidential to Editor” section, and submit your "Accept" recommendation.

Reviewer #4: All comments have been addressed

2. Is the manuscript technically sound, and do the data support the conclusions?

Reviewer #4: Yes

3. Has the statistical analysis been performed appropriately and rigorously? 

Reviewer #4: Yes

4. Have the authors made all data underlying the findings in their manuscript fully available?

Reviewer #4: Yes

5. Is the manuscript presented in an intelligible fashion and written in standard English?

Reviewer #4: Yes

6. Review Comments to the Author

Reviewer #4: In the manuscript titled “Prognostic value of SPARC in hepatocellular carcinoma: a systematic review and meta-analysis” the authors, have answered all the questions and edited according.

7. PLOS authors have the option to publish the peer review history of their article (what does this mean?). If published, this will include your full peer review and any attached files.

Reviewer #4: **Yes: **Ratnakar Tripathi

---

## [Editor Report · Acceptance letter]

10 Aug 2022

PONE-D-21-33504R2 

Prognostic value of SPARC in hepatocellular carcinoma: a systematic review and meta-analysis 

Dear Dr. Xia:

I'm pleased to inform you that your manuscript has been deemed suitable for publication in PLOS ONE. Congratulations! Your manuscript is now with our production department. 

Kind regards, 

on behalf of

Dr. Raj Kumar Koiri 

Academic Editor

PLOS ONE